ecology/genetics

badgers, Ireland, Britain, genetics, phylogeography, colonization

**Author for correspondence:**
Adrian Allen
e-mail: adrian.allen@afbini.gov.uk

# Genetic evidence further elucidates the history and extent of badger introductions from Great Britain into Ireland

Adrian Allen[1], Jimena Guerrero[2], Andrew Byrne[1], John Lavery[1], Eleanor Presho[1], Emily Courcier[3], James O'Keeffe[4], Ursula Fogarty[5], Richard Delahay[6], Gavin Wilson[7], Chris Newman[8], Christina Buesching[8], Matthew Silk[9], Denise O'Meara[10], Robin Skuce[1], Roman Biek[11] and Robbie A. McDonald[9]

[1]Agri-Food and Biosciences Institute, Belfast, UK
[2]Centre D'Ecologie Fonctionelle et Evolutive, Montpellier, France
[3]Department of Agriculture, Environment and Rural Affairs, Belfast, UK
[4]Department of Agriculture Food and the Marine, Ireland
[5]Irish Equine Centre, County Kildare, Ireland
[6]Animal and Plant Health Agency, UK
[7]Biocensus Ltd, Gloucestershire, UK
[8]Wildlife Conservation Research Unit, University of Oxford, UK
[9]Environment and Sustainability Institute, University of Exeter, Penryn, UK
[10]Waterford Institute of Technology, Ireland
[11]University of Glasgow, UK

 AA, 0000-0002-9077-8662; JG, 0000-0002-2197-2402;
MS, 0000-0002-8318-5383; RAM, 0000-0002-6922-3195

The colonization of Ireland by mammals has been the subject of extensive study using genetic methods and forms a central problem in understanding the phylogeography of European mammals after the Last Glacial Maximum. Ireland exhibits a depauperate mammal fauna relative to Great Britain and continental Europe, and a range of natural and anthropogenic processes have given rise to its modern fauna. Previous Europe-wide surveys of the European badger (*Meles meles*) have found conflicting microsatellite and mitochondrial DNA evidence in Irish populations, suggesting Irish badgers have arisen from admixture between human imported British and Scandinavian animals. The extent and history of contact

between British and Irish badger populations remains unclear. We use comprehensive genetic data from Great Britain and Ireland to demonstrate that badgers in Ireland's northeastern and southeastern counties are genetically similar to contemporary British populations. Simulation analyses suggest this admixed population arose in Ireland 600–700 (CI 100–2600) years before present most likely through introduction of British badgers by people. These findings add to our knowledge of the complex colonization history of Ireland by mammals and the central role of humans in facilitating it.

# 1. Background

The origins of Ireland's comparatively depauperate mammalian fauna are an ongoing area of research [1,2], which has revealed the varied and complex processes of colonization [3]. The colonization of native species such as stoats (*Mustela erminea*) [4] and Irish hares (*Lepidus timidus*) [5] is likely to predate the Last Glacial Maximum (LGM), 19–23 thousand years ago (kya) [3]. However, Ireland has been an island for the last 15 000 years [6], suggesting that natural colonization by non-volant animals is implausible after the LGM. Anthropogenic introductions, since the Mesolithic, have been proposed for a variety of species [3], with genetic studies finding the evidence of an Atlantic fringe signal in Irish mammals such as pine martens (*Martes martes*) [7] and pygmy shrews (*Sorex minutus*) [8], linking Irish animals to their counterparts in southwest Europe. However, for the pygmy shrew, it has been shown that while cytochrome *b* sequences linked extant Irish populations to those in Spain, the same sequence type was also found in Great Britain (GB), with further detailed microsatellite and Y chromosome typing indicating this was likely the historical source of origin for the extant Irish population [2]. The latter highlights the importance of sampling genetic diversity more widely, before ruling Britain out as a source for Irish mammals.

The European badger (*Meles meles*) exemplifies the challenges in understanding the biogeography of Ireland [9–12]. Previous microsatellite genotyping and mitochondrial DNA (mtDNA) analyses have shown that Irish badgers appear to have a mixed genetic heritage. mtDNA haplotypes are generally most similar to those currently observed in Scandinavia, leading to the hypothesis of Viking-aided introduction of animals into Ireland [12]. However, microsatellite data suggested a close relationship to British contemporaries [12]. The latter study, however, while useful for making broad historical inferences about glacial refugia, re-colonization of western Europe post LGM and highlighting potential origins of Irish badgers [12], used only 40 Irish samples from two small locales in the northeast and southeast of the island [12]. In this study, we make use of a previously assembled, island-wide sampling of Irish badgers [13] and genetically contrast it to a widespread sampling of GB badgers to more comprehensively inform on the extent, pathway and timing of genetic admixture between badgers from both islands.

# 2. Material and methods

## 2.1. Sample collection

A total of 545 badger muscle tissue samples were available for use in this study (full breakdown of submissions per region in electronic supplementary table S1). These were collected from the Republic of Ireland (RoI), Northern Ireland (NI) and GB. One hundred and seventy six badger carcasses from an ongoing road traffic accident (RTA) survey were collected by the Department of Agriculture, Environment and Rural Affairs Northern Ireland (DAERA-NI) across all six counties of NI during the period from September 2011 to March 2013. GPS locations of carcasses were logged and a tissue sample was stored for DNA extraction. 278 badger carcasses from ongoing bovine TB control-related culling operations in the RoI were collected by the Department of Agriculture, Food and the Marine (DAFM) during 2014. These badger carcasses were collected from sites distributed across 23 of the 26 counties of RoI. The three counties excluded were Donegal, Dublin and Louth. GPS coordinates of locations of culled animals were collected and a tissue sample stored for DNA extraction. A map of Ireland containing county names and boundaries is shown in electronic supplementary material, figure S1. For 21 samples across both territories, no geo-location data were available. Geo-locations for Irish animals are found in electronic supplementary material, data S1. The map in figure 1*a* illustrates the location of all Irish badgers sampled.

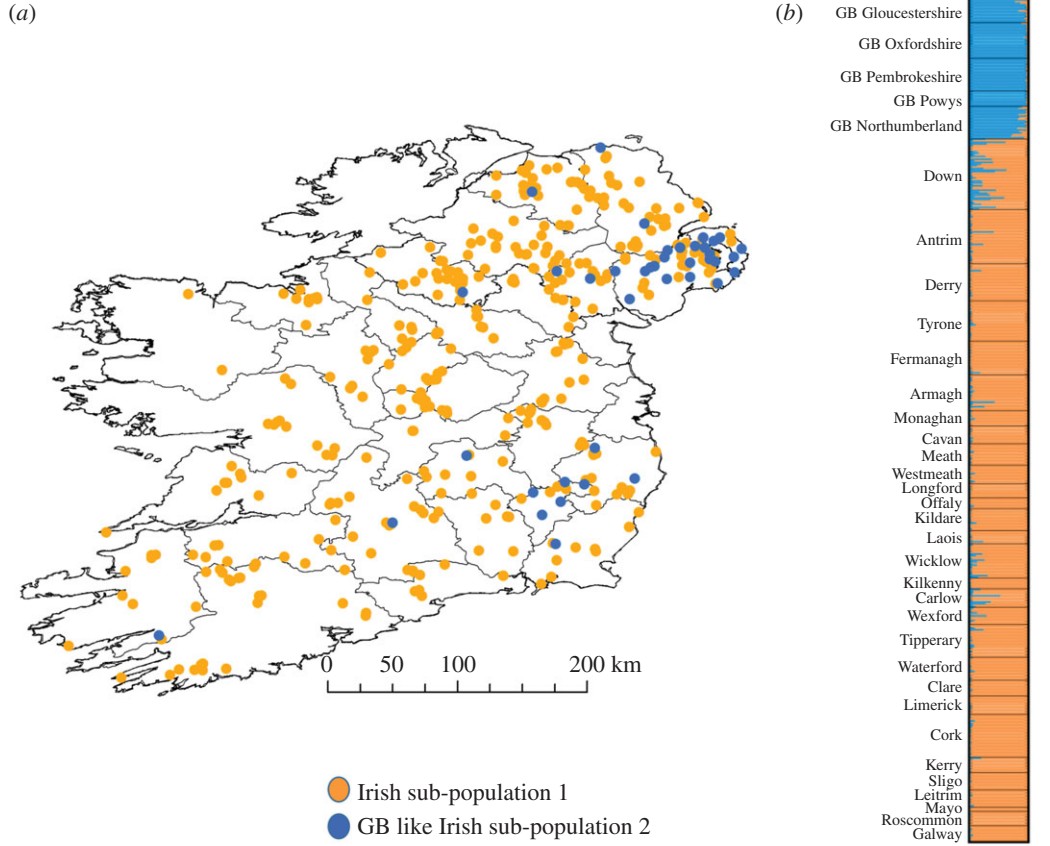

**Figure 1.** (*a*) spatial distributions of Irish sub-population 1 and Irish sub-population 2. (*b*) STRUCTURE microsatellite admixture bar-plot *K*=2 for all badgers.

Ninety-one additional badger hair samples were made available from five regions of GB—Gloucestershire, Oxfordshire, Pembrokeshire, Powys and Northumberland. No geo-location data were supplied for these animals.

## 2.2. DNA extraction, microsatellite genotyping and mitochondrial DNA sequencing

DNA extraction and microsatellite genotyping with a 14 loci panel derived from the work of Carpenter *et al.* [14] have previously been described [13].

All badgers were Sanger sequenced for a 214 bp section of the mitochondrial D loop. Specifically, a 214 bp section of the mtDNA control region was amplified using the primers LRCB1 5′-TGG TCT TGT AAA CCA AAA ATG – 3′ and H16498 5′ – CCT GAA GTA AGA ACC AGA – 3′. PCR conditions consisted of 1 cycle of 96°C for 15 min, followed by 40 cycles of 96°C/60 sec, 50°C/60 sec, 72°C/60 sec, followed by a final elongation at 72°C for 10 min [15]. Sequencing of PCR products was undertaken using the Life Technologies big dye terminator 3.1 kit (Life Technologies, Paisley, UK), following the manufacturer's instructions. Electrophoresis and data collection were performed on a Life Technologies ABI3130xl genetic analyser with final analysis conducted using Life Technologies Sequencing Analysis 5.2 software. All sequence data were aligned and compared to previously submitted mitochondrial control region data from *Meles meles* using the NCBI Basic Local Alignment Search Tool BLAST (http://blast.ncbi.nlm.nih.gov/). Haplotype identities were those defined by O'Meara *et al.* [10].

All genetic data were mapped using ArcGIS ArcMAP 10 using Latitude and Longitude coordinates based on the Irish Grid [16].

## 2.3. Standard population genetic indices

In GENEPOP [17], we assessed the number of alleles (Na), observed ($H_O$) and expected ($H_E$) heterozygosity and the inbreeding/fixation co-efficient ($F_{IS}$). We performed this for the combined GB

samples, the combined Irish samples, each GB region and the two STRUCTURE inferred Irish sub-populations (See Results).

Badgers from NI were sampled after road traffic accidents whereas their RoI contemporaries were sampled after sett side trapping, we wanted to rule out any possibility that behavioural differences potentially related to ranging could make direct genetic comparison inappropriate. We therefore compared allele frequency data between the County Down RTA population and another County Down population sampled sett side (data not shown).

## 2.4. STRUCTURE analysis parameters

We analysed microsatellite data from all badgers using STRUCTURE 2.3.4 to determine population sub-structure across both islands [18]. Since any potential sub-population's history in Ireland is *a priori* unknown and may result from the ancient or recent divergence of all sub-populations from a common ancestral population or human-aided introductions [10,12], we ran correlated and independent allele frequency, admixture models, assessing which produced the highest log likelihood for best fitting values of $K$. Both models were run without location prior. To infer best fitting number of $K$, we used the $\Delta K$ method of Evanno *et al.* [19] over consecutive values from $K = 1$ to $K = 10$ with a burn in of 50 000 and Markov chain length of 100 000, for 20 iterations per $K$ value. The convergence of key statistics along the burn in chain was assessed as per the STRUCTURE manual. We then analysed the data using STRUCTURE Harvester [20]. Data for each $K$ value ($n = 20$) were processed using the program CLUMPP [21], with final illustrations produced using DISTRUCT [22].

We quantified genetic differentiation between GB regional populations and the two Irish sub-populations inferred by STRUCTURE by calculating pairwise $F_{ST}$ values using FSTAT 2.9.3.2 [23]. Statistical significance of pairwise values was tested by 420 permutations with corrections for multiple comparisons. Genetic differentiation between populations was also quantified using Jost's D statistic [24] calculated by the mmod package [25] in the R environment v. 3.2.2 [26].

## 2.5. Discriminant analysis of principal components method

We also assessed sub-structure using the multivariate Discriminant Analysis of Principal Components (DAPC) method [27], which does not rely on maximizing linkage disequilibrium between loci and Hardy–Weinberg equilibrium. We performed DAPC in the adegenet package [28] in the R environment v. 3.2.2 [26]. The *find.clusters* function was used first to assign individual samples to proposed sub-populations. We retained all 80 principal components for this initial step. We then applied the DAPC analysis function to the number of clusters exhibiting the lowest Bayesian Information Criterion (BIC) to produce a scatterplot, retaining 40 principal components which accounted for 90% of the observed variance, and all linear discriminants.

## 2.6. Do it yourself approximate Bayesian computation analysis parameters and historical model details

To determine the timing of the contact between Irish and British badgers, we implemented an Approximate Bayesian Computation (ABC) on microsatellite alone and combined microsatellite and mtDNA data in the software package do it yourself approximate Bayesian computation (DIYABC) [29] for a sub-set of Irish badgers that exhibited most admixture with British badgers (see below). British badger samples were treated as a separate population, and compared to the two Irish sub-populations ($n = 414$ & $n = 40$). Mutation rates were retained at the default setting as per the DIYABC manual. After initial exploratory analyses, the following parameters were set. Effective population sizes for the two Irish sub-populations were allowed to vary between 10 and 10 000 individuals, and 10 and 15 000 for the GB population. Two historical scenarios were considered. Both are visually represented in figure 2. In scenario 1, British badgers (Population 1) and the proposed original Irish badger population (Population 2) split from an ancestral population at t2, 500–15 000 generations before present (gbp) (3000 to 90 000 years before present to take in the period of the LGM approximately 20 000 ybp). Then at t1, 10–500 gbp (60–3000 ybp to encompass recent history including the colonization of Ireland from Britain from the Mesolithic to modern times) a new, and more British-like, Irish sub-population (Population 3) split from the original Irish population. In scenario 2, British badgers (Population 1) and the proposed original Irish badger population (Population 2) split

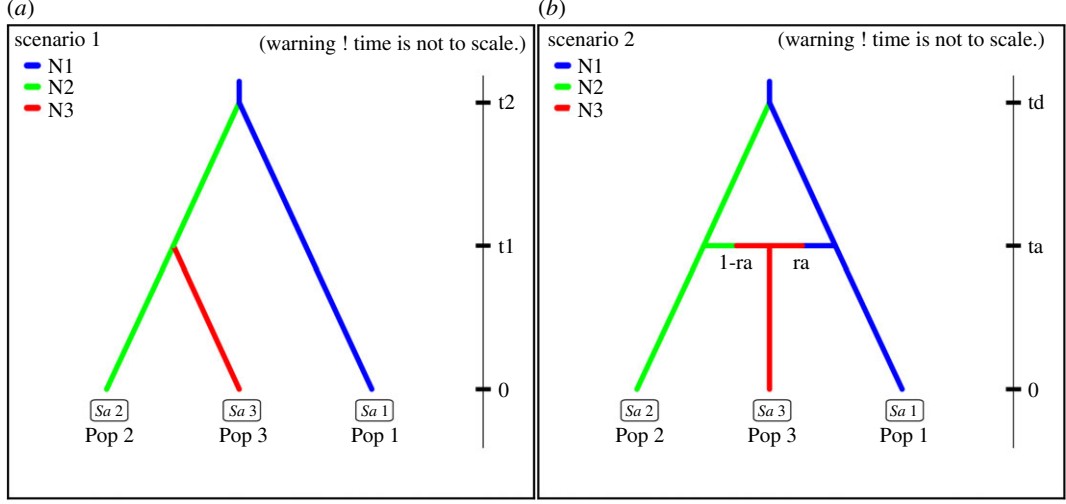

**Figure 2.** DIYABC simulated historical scenarios: Pop 1 = GB badger population; Pop 2 = Irish badger sub-population 1 exhibiting less than 15% GB like genetic heritage; Pop 3 = Irish badger sub-population 2 exhibiting more than 15% GB like genetic heritage. (a) Scenario 1–divergence of second Irish sub-population from original Irish sub-population; (b) Scenario 2–admixture event between British and Irish badgers resulting in emergence of GB like Irish sub-population 2.

from an ancestral population at td, 500–15 000 generations before present (gbp). Then, at ta, 10–500 gbp, an admixture event between the first two populations brought about the emergence of a third, British-like, Irish population (Population 3). Each scenario was considered equally probable. For each population, the following summary statistics were calculated for microsatellite loci—mean number of alleles and mean genic diversity, Garza and Williamson's M-ratio, $F_{st}$ and $\delta\mu^2$ distance. One million datasets were simulated per scenario. Summary statistics calculated for the mtDNA data were number of haplotypes, number of segregating sites, mean of pairwise differences, variance of pairwise differences and pairwise $F_{st}$. We assessed the reliability of the simulated data in comparison to observed prior and posterior data by PCA. DIYABC produced posterior probabilities of both scenarios and posterior distributions of historical parameters by logistic regression using the closest 1% of simulated datasets to the observed data.

# 3. Results

## 3.1. Lower genetic diversity and weaker population structure in Irish badgers compared to Britain

Z tests of allele frequencies compared between RTA animals and sett side capture badgers from County Down, when corrected for multiple comparisons indicated there were no significant allele frequency differences, suggesting there is no representative bias in the NI RTA dataset compared to RoI animals.

Standard population genetic indices of badgers across Britain and Ireland are shown in table 1. Combined British and Irish populations (table 1a and b) exhibit similar average numbers of alleles per locus. Irish badgers exhibited lower genetic diversity (expected (He) and observed (Ho) heterozygosity, 0.56 and 0.48, respectively) and fixation indices ($F_{is}$ 0.14) than their British contemporaries (He 0.67 and Ho 0.53; $F_{is}$ 0.22), owing to higher relative frequencies of multiple alleles in GB animals (data not shown). As a result, GB sub-populations exhibited greater sub-population genotype differentiation compared to that of Ireland.

## 3.2. Population sub-structure–links between British and Irish badgers

STRUCTURE analysis indicated that a sub-set of Irish badgers exhibited the evidence of historical admixture with British badgers (figure 1). The correlated allele frequency model Evanno $\Delta K$ plot exhibited a peak at $K = 2$ and a smaller one at $K = 6$ (electronic supplementary material, figure S2), with a higher mean log likelihood than the independent allele frequencies model (electronic

**Table 1.** Population genetic summary statistics averaged across all 14 loci for badgers from A, Great Britain; B, Ireland; C, Gloucestershire; D, Oxfordshire; E, Pembrokeshire; F, Powys; G, Northumberland; H, Irish sub-population 1; I, British-like Irish sub-population 2. N, number of individual badgers; Na, number of alleles observed at each locus; He, expected heterozygosity; Ho, observed heterozygosity; Fis Fixation index (Weir and Cockerham)–inbreeding of individuals relative to population.

|   |    | mean |
|---|----|------|
| A | N   | 90.4 |
|   | Na  | 5.6  |
|   | He  | 0.67 |
|   | Ho  | 0.53 |
|   | Fis | 0.22 |
| B | N   | 453.2 |
|   | Na  | 5.9  |
|   | He  | 0.56 |
|   | Ho  | 0.48 |
|   | Fis | 0.14 |
| C | N   | 16   |
|   | Na  | 4.3  |
|   | He  | 0.65 |
|   | Ho  | 0.62 |
|   | Fis | 0.09 |
| D | N   | 23   |
|   | Na  | 3.9  |
|   | He  | 0.54 |
|   | Ho  | 0.53 |
|   | Fis | 0.02 |
| E | N   | 21   |
|   | Na  | 3.6  |
|   | He  | 0.53 |
|   | Ho  | 0.43 |
|   | Fis | 0.17 |
| F | N   | 10   |
|   | Na  | 3.1  |
|   | He  | 0.54 |
|   | Ho  | 0.47 |
|   | Fis | 0.11 |
| G | N   | 20.4 |
|   | Na  | 4.1  |
|   | He  | 0.58 |
|   | Ho  | 0.57 |
|   | Fis | 0.03 |
| H | N   | 413.3 |
|   | Na  | 5.8  |
|   | He  | 0.55 |
|   | Ho  | 0.48 |
|   | Fis | 0.13 |
| I | N   | 40   |
|   | Na  | 5.1  |
|   | He  | 0.64 |
|   | Ho  | 0.56 |
|   | Fis | 0.12 |

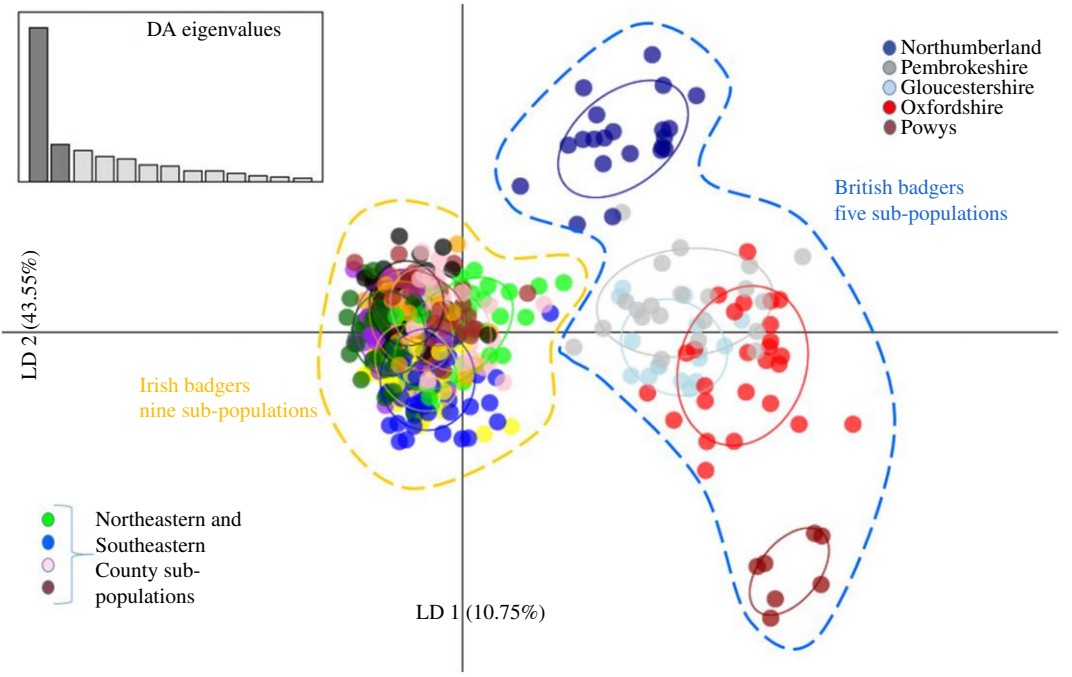

**Figure 3.** DAPC plot—multivariate genetic analysis of all Irish and GB badgers and assignment to sub-population clusters.

supplementary material, table S2). The $K = 2$ data indicated that Irish badgers in Counties Down, Antrim, Armagh, Laois, Wicklow, Carlow, Wexford and Tipperary exhibited a British-like genetic heritage (figure 1). The correlated allele frequencies STRUCTURE model at $K = 6$ separated the GB and Ireland meta-population into two GB sub-populations and four Irish sub-populations. The GB sub-populations were split into northern and southern groups—sub-population one was comprised of badgers from Gloucestershire, Oxfordshire, Pembrokeshire and Powys whilst sub-population two was made up exclusively of badgers from Northumberland (data not shown). The $K = 6$ analysis output also indicated that while there was a significant admixture apparent between the four badger sub-populations, the signal of GB genetic heritage was still apparent in the northeastern County Down and southeastern Counties of Wicklow and Carlow (electronic supplementary material, figure S3). Given that both hierarchical STRUCTURE models supported the GB genetic heritage in these regions, in line with the objective in this study to investigate links between British badgers and Irish contemporaries, we focused our efforts on the $K = 2$ analysis output data, which exhibited the strongest statistical support (electronic supplementary material, table S2). Using a previously described population assignment Q score threshold of 85% for this Irish badger dataset [13], we defined a 'native' Irish sub-population, Irish sub-population 1, accounting for 414 of the surveyed Irish badgers (figure 1a). The remaining 40 badgers, having at least 15% British heritage, were designated as Irish sub-population 2. The sub-populations currently inhabit distinct geographic locations, with sub-population 2 found primarily in the northeastern and southeastern counties of Ireland (figure 1a). Both Irish sub-populations and all British regional sub-populations exhibited significant pairwise genetic differentiation from one another, as measured by $F_{st}$ (electronic supplementary material, table S3). British regional sub-populations were least differentiated from Irish sub-population 2 (Average $F_{st} = 0.17$; Average Jost's $D = 0.32$). Specifically, badgers from Gloucestershire and Northumberland were the British populations most similar to Irish sub-population 2 ($F_{st} = 0.15$). For the DAPC cluster analysis, the *find.clusters* function, executed in adegenet returned a curve which indicated that $K = 14$ had the lowest BIC (electronic supplementary material, figure S4). The resulting $K = 14$ DAPC scatterplot is shown in figure 3 with Linear Discriminant axis 1 accounting for 10.75% of the observed variance and Linear Discriminant axis 2, 43.55% of the variance. Irish badgers were assigned to nine tightly clustered sub-populations (figure 3). The five British regional locations sampled were assigned to five distinct regional clusters (figure 3). All British badgers were distinctly clustered apart from the Irish meta-cluster; however, some members of Irish clusters primarily from northeastern and southeastern counties were intermediate in genetic space between the Irish meta-cluster and the British sub-population clusters found in Gloucestershire and

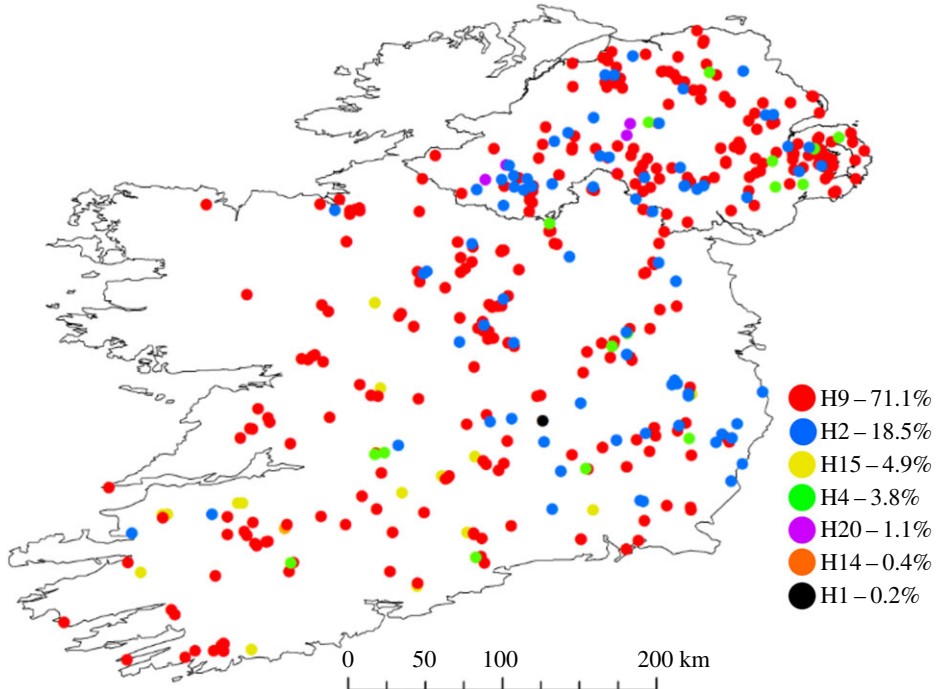

**Figure 4.** Spatial distribution of mitochondrial DNA haplotypes of badgers in Ireland and their percentage abundance.

Pembrokeshire. Of the 40 Irish badgers from STRUCTURE sub-population 2, 31 were assigned to the DAPC clusters from northeastern and southeastern counties.

Seven mtDNA haplotypes were identified in the Irish badger population. Haplotypes H2, H9, H14, H15 and H20 accounted for 96.0% of all haplotypes in Ireland. Five haplotypes, H2, H9, H14 and H15 had previously been reported by O'Meara *et al.* [10]. Previously, the H9 haplotype, the most common haplotype in Ireland (71.1%), was found to be common in Scandinavia, and in low numbers in Spain, Central Europe and Scotland [10]. H2, the next most common Irish haplotype (18.5%), has previously been observed in Central Europe [10], while H15 (4.9%) has previously been observed in the Iberian Peninsula [10]. H20 (0.2%) was closely related to the H9 haplotype, differing at only one base and has not been seen in other territories to date. The H14 haplotype (0.4%) was found, as before, in the southwest of Ireland [10]. Three haplotypes were identified in the British badger population. H1 and H4 made up the majority of the isolates (57.1% and 40.6%, respectively). H1 was found across all sampled British counties, while H4 was found in all counties except Northumberland. The Scandinavian like H9 haplotype was least common (2.2%) and was found only in Northumberland. Haplotypes H4 and H1 accounted for 3.8% and 0.2%, respectively, of the observed mtDNA diversity in Ireland, and while not observed to cluster in tight foci in the northeastern and southeastern counties of Ireland (figure 4), two distinct 95% kernel densities with their centroids focused on these regions were apparent in the data (electronic supplementary material, figure S5). Our mtDNA data is not directly comparable to that of Frantz *et al.* on account of our sequencing of a shorter region of the mitochondrial control region. However, as with O'Meara *et al.* [10], we suggest that the majority of Irish badgers exhibit an mtDNA heritage consistent with colonization from countries from the Atlantic fringe—Iberia and Scandinavia. Animals with this heritage are widely spread across the island (figure 4).

## 3.3. Do it yourself approximate Bayesian computation inference of badger sub-population history

The combined microsatellite and mtDNA data did not yield satisfactory support for either of the two historical scenarios simulated in DIYABC. PCA evaluation indicated neither historical simulation model explained the observed data (not shown). Similar lack of support for simulated historical models has been observed before when using combined mtDNA and microsatellite data to address the effects of admixture in forming extant badger populations in Europe [12]. Additionally, in other species which colonized Ireland post LGM, sole use of microsatellites for DIYABC inference of

**Table 2.** DIYABC historical parameter estimates for admixture scenario 2. Extant effective population sizes are in absolute numbers. All time parameters are expressed as generations before present (gbp).

| parameter | mean | median | 95% CI |
|---|---|---|---|
| N1 – Eff. Pop. size British population (n) | 6710 | 6480 | 2160–12 900 |
| N2 – Eff. Pop. size Irish sub-pop 1 (n) | 4240 | 4070 | 1300–8410 |
| N3 – Eff. Pop. size Irish sub-pop 2 (n) | 5530 | 5510 | 1380–9650 |
| ta – time of admixture (gbp) | 145 | 110 | 17–436 |
| ra – rate of admixture from British population | 0.25 | 0.23 | 0.05–0.56 |
| td – time of divergence pops 1 and 2 (gbp) | 1820 | 1350 | 596–6290 |

population history has been preferred owing to the observation that mtDNA variation likely reflects more ancient, late Pleistocene divergence, while microsatellites better reflect more recent post glacial divergence and admixture/gene-flow [30].

Microsatellite data alone did exhibit greater support for historical scenarios, so from here-on, we refer to the results derived from that dataset. A higher posterior probability for DIYABC scenario 2 (0.65 CI 0.63–0.66) versus scenario 1 (0.35 CI 0.34–0.37). PCA evaluation of the DIYABC analysis for scenario 2 indicated the chosen historical model explains the observed data (electronic supplementary material, figure S6). In keeping with the DIYABC manual guidelines, observed data are seen to cluster in the centre of simulated posterior datasets, which are themselves nested within simulated prior datasets. Historical parameter estimates from DIYABC are shown in table 2. Data indicated that the appearance of Irish sub-population 2 was more likely due to an admixture event occurring in the recent past—110 generations before present (CI 17–436 gbp). Given contemporary badger generation time of approximately six years [31], this suggests the admixture event occurred 600–700 (CI 100–2600) years before present (ybp). Data also suggested that the divergence of the two European populations that went on to colonize Britain and Ireland occurred 1820 generations before present (CI 596–6290). Again, given a badger generation time of approximately six years, this suggests this event occurred 10 920 ybp (CI 3576–37 740).

Genotype and mtDNA sequence type data are available in electronic supplementary material, Data S1 along with additional detail. The genetic contribution from British badgers in founding Irish sub-population 2 was estimated to be 23%, with admixture from Irish sub-population 1 accounting for the remaining 77%.

# 4. Discussion

We have shown that a sub-set of Irish badgers, primarily located in the northeast and southeast of the island, share genetic ties, both nuclear and mitochondrial, with British contemporaries, most likely as a result of colonization from GB in the recent past, involving a small number of animals which admixed with already resident Irish animals. This finding expands upon prior evidence for badger colonization of Ireland, highlighting added complexity by showing how the extant Irish badger population has been formed by separate, region specific, colonization processes, from multiple sources and subsequent admixture.

There has been justified recent criticism of an over reliance on the outputs from STRUCTURE for inferring population admixture and sub-structure [32], with the acknowledgement that there is a bias towards $K = 2$ appearing in analyses of these types. $K = 2$ findings can be a spurious finding, therefore, with artefacts in admixture analyses introduced by coercion, leading to inaccurate assumptions of genetic heritage. However, we believe this is unlikely to be the case in this instance. When one considers the full range of evidence we present, we believe that the chances such an artefact has arisen are low. At both hierarchical levels identified ($K = 2$ and $K = 6$), the STRUCTURE data is congruent, suggesting GB like genetic heritage in badgers from southeastern and northeastern counties. If an artefact driven by some stochastic effect of the STRUCTURE algorithm was the reason for our data, why would its effects only be limited to Irish animals from the northeastern and southeastern counties, the very regions Frantz et al.'s [12] independent study shows contain animals whose nuclear genomes share similarities with British contemporaries? In addition, why would these

same areas be the centroids for mtDNA haplotypes associated with British animals? Why would an independent DAPC analysis point to the same conclusion? The weight of evidence we present suggests our STRUCTURE analysis is finding genuine evidence of admixture with British animals.

Our data are congruent with the role of human-aided animal introductions to Ireland [3] given that we demonstrate the Irish sub-population with genetic links to GB likely arose 600–700 ybp through an admixture event. Since Ireland is thought to have become an island 15 000 ybp [6], this admixture event was probably facilitated by translocation of badgers by people from western GB to eastern Ireland. An anecdotal, but interesting observation, is that the areas of Ireland exhibiting the most admixture with British badgers, were also the regions that had the most evidence of human genetic admixture from Britain [33], and the commonest occurrence of British surnames [34]. Mass colonization of Ireland from Britain occurred in the Norman and Elizabethan eras (12th and 16th centuries), a time frame consistent with the findings of the DIYABC analysis.

Our data also help to further understand the findings of Frantz *et al.* [12], who reported Irish badgers as exhibiting mixed genetic heritage—Scandinavian like mtDNA haplotypes, but GB like microsatellites. While we sequenced a shorter mtDNA fragment which does not provide the highest phylogeographic resolution, our data suggesting the Irish population exhibits a primarily Atlantic fringe heritage is broadly congruent with Frantz *et al.* [12], whose analysis of a longer fragment found the most common mtDNA haplotypes in Ireland are those of Scandinavian origin, consistent with the hypothesis of a role for Viking involvement in transportation of badgers into Ireland. However, we suggest the reason Frantz *et al.* [12] observed the admixed heritage of Irish badgers is due to the fact they only surveyed in two small areas of the northeast and southeast of the island—the regions, from our analyses, whose badger populations exhibit closest ties to British contemporaries, but also areas in which the H9 Atlantic fringe/Scandinavian haplotype is commonly found. Our island-wide sampling suggests that admixture between British and previously resident Irish badgers, predominantly of an Atlantic fringe origin, is not a general feature of the Irish badger population, but rather a feature of animals on the eastern coast. We also differ in the relative timings of the colonization events from the Atlantic fringe countries and Britain. Frantz *et al.* [12] have suggested on the basis of shared microsatellite and mtDNA data alone that colonization from Britain preceded Viking-aided incursions of additional animals. However, it seems possible from our findings that badgers of an Atlantic fringe origin may already have been present before the incursion from Britain. The island-wide distribution of the Atlantic fringe haplotypes compared to the east coast localization of the smaller number of British-associated haplotypes alongside congruent microsatellite data suggests an incursion from GB of a comparatively small number of animals, which did not disperse very far. DIYABC analyses support this hypothesis both in suggesting the admixture event occurred around 700 years before present, a time period after the Viking invasions, and in suggesting an already resident Irish badger population made a considerably larger genetic contribution to the admixed population, than animals from GB.

A significant caveat to our hypothesis as detailed above is that the period of time identified by DIYABC exhibits a wide confidence interval, suggesting this translocation took place at some point between the late Bronze Age and the early twentieth century. There is considerable evidence for human-aided movement of animals into Ireland during this period. Red foxes (*Vulpes vulpes*) and pine martens were introduced during the Bronze (2500–500 BC) and Iron Ages (500 BC – 400 AD) [3], presumably as food and fur resources [3]. The Vikings (8th–10th centuries) played a role in translocating badgers from their homelands to Ireland [12], with wider evidence pointing to their movement of animal species around European coastlines [3]. Norman settlement of Ireland (12th century) is associated with the introduction of hedgehogs (*Erinaceus europaeus*), rabbits (*Oryctolagus cuniculus*), red squirrels (*Sciurus vulgaris*) and fallow deer (*Dama dama*) to the island [3], likely as food species. Any of these anthropogenic mechanisms could account for the translocation we have described, but it is difficult to be precise given the data. Although uncertainty around this date estimate was substantial, the limited spatial extent of admixed badger populations in Ireland is potentially consistent with a more recent introduction.

In addition, our DIYABC data suggest that the two European sub-populations that went on to found the modern populations of Britain and Ireland post LGM (populations 1 and 2 in our DIYABC analysis), diverged around 11 000 ybp. This timing predates the transition of Britain from being a European peninsula to becoming an island around 8500 ybp [35]. Also, the number of generations before present (1820 CI 596–6290) is very similar to that elucidated by Frantz *et al.* (1760 CI 525–6390) for the occurrence of the admixture event between Iberian and Balkan refugial badgers that founded the modern day Scandinavian population [12]. We suggest therefore that around the same time the

modern day Scandinavian population was being formed by admixture somewhere in Europe, another European sub-population diverged that went on to colonize Britain, becoming separated from continental contemporaries as Britain became an island. Animals from the Atlantic fringe then went on, through human aid, to found the population in Ireland.

In summary, these data further advance our knowledge of the history of colonization of Ireland by mammals. They also help to deepen our understanding of the extant Irish badger population structure and the forces that shaped it, providing general lessons about the interaction of environmental and anthropogenic factors in determining the phylogeography of European mammals. Future work to undertake microsatellite comparison of the whole Irish population to the wider European population would be beneficial in further elucidating the finer details of how and when Ireland was colonized by this important species. Comparison to Scandinavian badgers would perhaps be most useful given how common the Atlantic fringe haplotypes are in Ireland; however, our finding that the second most common mtDNA haplotype (H2 – 18.5%) is one commonly seen in Central Europe [10] suggests wider comparison may reveal yet more complexity in the colonization process.

Ethics. All Irish samples were collected incidentally alongside ongoing government efforts to eradicate bovine tuberculosis under the relevant legislation of each territory - NI: Tuberculosis Control order (NI) 1999 No.263; Tuberculosis Scheme order (NI) 1999 No.264); RoI: Culling of badgers is carried out under licence by the National Parks and Wildlife Service, with powers conferred by the Wildlife Acts 1976 to 2010.

Badger samples from GB were collected under Home Office and Natural England licenses.

Data accessibility. Data are available as electronic supplementary material, data S1.

Authors' contributions. A.A., J.G., A.B., R.S. and R.B. conceived the study and wrote the manuscript. A.A., E.P. and J.L. performed laboratory work. A.A. performed analyses. Other authors supplied samples and drafted the manuscript.

Competing interests. We declare we have no conflict of interest.

Funding. This project was funded by the Department of Agriculture Environment and Rural Affairs, NI (DAERA-NI). Fieldwork in GB was funded by NERC (grant no. NE/M004546/1).

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
