## [Reviewer comments · Royal Society Open Science]

Review History

Decision letter (RSOS-200288.R0)

02-Mar-2020

Dear Dr Allen:

It is a pleasure to accept your manuscript entitled "Genetic evidence further elucidates the history and extent of badger introductions from Great Britain into Ireland." in its current form for publication in Royal Society Open Science.

Following your final, appropriate minor revisions to the paper in response to my acceptance of your appeal, we are pleased to be able to publish your article.

You can expect to receive a proof of your article in the near future. Please contact the editorial

Reports © 2020 The Reviewers; Decision Letters © 2020 The Reviewers and Editors; Responses © 2020 The Reviewers, Editors and Authors. Published by the Royal Society under the terms of the Creative Commons Attribution License <http://creativecommons.org/licenses/by/4.0/>, which permits unrestricted use, provided the original author and source are credited

office (openscience_proofs@royalsociety.org) and the production office (openscience@royalsociety.org) to let us know if you are likely to be away from e-mail contact -- if you are going to be away, please nominate a co-author (if available) to manage the proofing process, and ensure they are copied into your email to the journal.

on behalf of Dr Steve Brown (Associate Editor) and Dr Steve Brown (Subject Editor).
